# A DVD-MoS_2_/Ag_2_S/Ag Nanocomposite Thiol-Conjugated with Porphyrins for an Enhanced Light-Mediated Hydrogen Evolution Reaction

**DOI:** 10.3390/nano10071266

**Published:** 2020-06-29

**Authors:** Leonardo Girardi, Matías Blanco, Stefano Agnoli, Gian Andrea Rizzi, Gaetano Granozzi

**Affiliations:** 1Department of Chemical Sciences and INSTM Unit, University of Padova, Via F. Marzolo 1, 35131 Padova, Italy; leonardo.girardi@phd.unipd.it (L.G.); stefano.agnoli@unipd.it (S.A.); gaetano.granozzi@unipd.it (G.G.); 2Department of Organic Chemistry, Universidad Autónoma de Madrid, Ciudad Universitaria de Cantoblanco, Calle Francisco Tomás y Valiente, 7, 28049 Madrid, Spain

**Keywords:** TMDC, MoS_2_-Ag_2_S, DVD, porphyrins, HER, proton relay, p-n junction

## Abstract

We have recently demonstrated in a previous work an appreciable photoelectrocatalytic (PEC) behavior towards hydrogen evolution reaction (HER) of a MoS_2_/Ag_2_S/Ag nanocomposite electrochemically deposited on a commercial writable Digital Versatile Disc (DVD), consisting therefore on an interesting strategy to convert a common waster product in an added-value material. Herein, we present the conjugation of this MoS_2_/Ag_2_S/Ag-DVD nanocomposite with thiol-terminated tetraphenylporphyrins, taking advantage of the grafting of thiol groups through covalent S-S bridges, for integrating the well-known porphyrins photoactivity into the nanocomposite. Moreover, we employ two thiol-terminated porphyrins with different hydrophilicity, demonstrating that they either suppress or improve the PEC-HER performance of the overall hybrid, as a function of the molecule polarity, sustaining the concept of a local proton relay. Actually, the active polar porphyrin—MoS_2_/Ag_2_S/Ag-DVD hybrid material presented, when illuminated, a better HER performance, compared to the pristine nanocomposite, since the porphyrin may inject photoelectrons in the conduction band of the semiconductors at the formed heterojunction, presenting also a stable operational behavior during overnight chopped light chronoamperometric measurement, thanks to the robust bond created.

## 1. Introduction

Transition metal dichalcogenides (TMDCs) are currently considered the flagship structures in the “beyond graphene realm” of material science research and development [1,2]. Since the 1960s, MoS_2_, and its related materials, have found application as lubricants, due to their relatively easy exfoliation, as (opto)electronic devices when the rather-perfect crystals are restricted to the nano-regime and/or as (photo)electrochemical (PEC) hydrogen evolution reaction (HER) catalysts [3,4,5,6,7], especially in the amorphous form [8]. Indeed, the HER is of capital importance for the energetic transition from the fossil-based fuels to a more sustainable energetic system [9]. Therefore, the scientific community is currently undertaking huge efforts to exploit the excellent properties of TMDCs for sustainable PEC-HER catalysts [7] to replace the expensive and critical materials based on platinum-group metals traditional catalysts [10]. Moreover, the fabrication of the electrode should respect a sustainable and circular approach. A very interesting strategy consists of using waste or products at the end-of-life and to convert them in added-value goods. Very recently, our research group described the preparation of an amorphous MoS_2_ based PEC-HER catalyst with excellent performance, employing, as a starting material, a recycled Digital Versatile Disc (DVD) [1]. The strategy consisted of the short potentiostatic electroreduction of (NH_4_)_2_MoS_4_ (typically a 3–4 min long procedure), using the clean nano-patterned 1 µm thick Ag layer of the DVD as electrode. This methodology allowed the in situ formation of amorphous MoS_2_ over the DVD surface, while releasing, at the same time, HS^−^ anions which react with the metallic Ag [11], forming a heterostructure composed of MoS_2_ and Ag_2_S (hereafter MoS_2_/Ag_2_S/Ag-DVD). The two sulphides produced a composite with a type II p-n junction on a highly conductive material like Ag that is able to boost the PEC-HER activity [12].

The MoS_2_ PEC-HER activity could also be increased by employing photosensitizers. Among the extended library of photoactive molecular systems, porphyrins are a class of particular organic architectures, with an 18 π-electron aromatic structure that presents a strong optical absorption in the visible spectrum, comprising the intense Soret (typically λ = 400–450 nm) and Q (λ = 550–650 nm) bands [13]. Interestingly, there is a wide gamut of well-established protocols that allow the modification of the optical and hydrophilicity properties of these macrocycles [14,15,16,17,18]. In addition, porphyrins are photochemically stable molecules whose redox potentials lie within electrochemical windows that are compatible for interesting processes [19], e.g., the HER one. Therefore, porphyrins have been widely investigated, since they can drive and promote charge-transfer phenomena [20,21,22,23,24]. Recently, we described how thiol-terminated porphyrins can be covalently conjugated with the edges of 2D-MoSe_2_ nanoflakes and generate a heterojunction that enhances the HER catalytic activity under light irradiation [25]. The same thiol conjugation chemistry should work on the low coordination sites present in the amorphous structure of MoS_2_ at the MoS_2_/Ag_2_S/Ag-DVD nanocomposite, hence, crafting a porphyrin modified material with improved performance in the PEC-HER.

Thus, in this work we present the thiol-terminated porphyrin decoration of the MoS_2_/Ag_2_S/Ag-DVD nanocomposite containing amorphous MoS_2_ and Ag_2_S. Simple dipping methodologies were employed to conjugate the pristine nanocomposite, employing hydrophilic (6,6′,6″,6‴-(porphyrin-5,10,15,20-tetrayl) tetrakis (3-(3-mercaptopropoxy)phenol), molecule **1**) and hydrophobic (3,3′,3″,3‴-((porphyrin-5,10,15,20-tetrayltetrakis (benzene-4,1-diyl)) tetrakis (oxy)) tetrakis (propane-1-thiol), molecule **2**) ad hoc synthetized porphyrins. The successful porphyrin-TMDC conjugation was tested by means of several characterization techniques (X-ray photoelectron spectroscopy (XPS), Fourier Transformed Infra-Red (FTIR) and Raman spectroscopies, scanning electron microscopy (SEM)), and the formation of S-S bridges between the molecule and the material, without modifying the morphology of the surface, has been fully demonstrated. In addition, different behavior is observed when the PEC-HER activity was tested with the two different final catalysts. While the sample conjugated with the hydrophobic porphyrin **2** (hereafter **2-**MoS_2_/Ag_2_S/Ag-DVD) shows a strong suppression of the HER activity (which is typical of functionalized MoS_2_ systems reported in the literature [26,27]), the hydrophilic sample **1-**MoS_2_/Ag_2_S/Ag-DVD provides a similar response under dark conditions compared to the pristine nanocomposite. However, improvements in the onset potential necessary to a current of 10 mA cm^−2^ are observed under light irradiation. Moreover, the hydrogen production is stable for at least 24 h under chopped light as a consequence of the covalent bond established between the light sensitizer and the active material.

## 2. Materials and Methods

All chemicals (reagents, solvents, acids, bases, silica gel and thin-layer chromatography plates) were supplied by Sigma Aldrich and used without further purification, if not otherwise stated. All air-sensitive reactions were carried out under N_2_ atmosphere employing Schlenk techniques with N_2_ degassed solvents, if not otherwise stated. High purity Milli-Q water (with a resistivity of 18.2 MΩ·cm) was employed in all the electrochemical tests, while deionized water (DI) was used during the organic synthetic protocols, when needed.

### 2.1. Synthetic Protocols

The synthesis of the nanocomposite MoS_2_/Ag_2_S/Ag-DVD consisted in the potentiostatic reduction of (NH_4_)_2_MoS_4_ (50 mM solution) using a conventional three electrode cell [1]. A standard Ag/AgCl electrode equipped with a KCl saturated gel (to avoid the diffusion of S^2-^ towards Ag) was used as reference, and a Pt wire was used as a counter electrode. The deposition was performed in a N_2_ purged solution, setting the potentials at −1.0 V (vs. Ag/AgCl) for 30 s. Control samples were also prepared. On the one hand, MoS_2_ was electrodeposited on a fluoride-doped tin oxide (FTO) slice (sample MoS_2_/FTO) using the same procedure as the DVD general preparation. On the other hand, a DVD sample without MoS_2_ was yielded by dipping the pristine DVD in a 30 mM water solution of Na_2_S (sample Ag_2_S/Ag) for 4 min. The synthesis of the porphyrins **1** and **2** (Scheme 1) was achieved by adapting the general Lindsey-Rothemund strategy [14,15,16,18,28], as described previously [25]. The spectroscopic data of the organic products (^1^H, ^13^C-NMR and UV-Vis spectroscopies) fully matched the reported data.

The modification of the MoS_2_/Ag_2_S/Ag-DVD material with the organic macrocycles was performed by incubating a 2 × 5 cm piece of the DVD containing the nanocomposite in a 0.02 mM ethanol solution of the molecule **1** or **2** for 2 days at room temperature (rt), and in the dark (see Figure 1). After this time, the piece was withdrawn from the solution and profusely rinsed with ethanol (6 × 25 mL), water (6 × 25 mL) and ethanol again (6 × 25 mL). Drying by blowing N_2_ afforded the modified materials **1-**MoS_2_/Ag_2_S/Ag-DVD and **2-**MoS_2_/Ag_2_S/Ag-DVD, respectively. Control experiments with samples Ag_2_S/Ag and MoS_2_/FTO were submitted to the conjugation procedure under the same experimental conditions.

### 2.2. Physicochemical Characterizations

The surface chemical characterization of the catalysts has been carried out using X-ray photoelectron spectroscopy (XPS) in a custom-made ultra-high vacuum (UHV) system, working at a base pressure of 10^−10^ mbar, equipped with an Omicron EA150 electron analyser and an Omicron DAR 400 X-ray source with a dual Al-Mg anode. Core level photoemission spectra (C 1s, N 1s, O 1s, Mo 3d and S 2p) were collected at rt with a non-monochromated Mg Kα X-ray source (1253.3 eV) and using an energy step of 0.1 zeV, 0.5 s of integration time, and a 20 eV pass energy. The microscale morphology of the materials was studied by scanning electron microscopy (SEM). SEM micrographs were acquired with a field emission source and a GEMINI column (Zeiss Supra VP35), working with an acceleration voltage of 5 or 10 kV, using in-lens high-resolution detection. The Raman spectra were collected using a ThermoFisher DXR Raman microscope, using a laser with an excitation wavelength of 532 nm (0.1 mW), focused on the sample with a 50 × objective (Olympus). Attenuated total reflectance infrared (ATR-IR) spectra were recorded with a Nicolet FT-IR (Thermo Optek, Rome, Italy) spectrometer. The nuclear magnetic resonance (NMR) spectra were recorded on a Bruker Advance 300 MHz (300.1 MHz for ^1^H, 75 MHz for ^13^C, 298 K); chemical shifts (*δ*) are reported in units of parts per million (ppm), relative to the residual solvent signals, and coupling constants (*J*) are expressed in Hz.

### 2.3. Photoelectrochemical Hydrogen Evolution Reaction Tests

The EC- and PEC-HER measurements were carried out in a custom designed three-electrode configuration cell using an Autolab PGSTAT-204 (Metrohm) potenziostat. A graphite rod was used as a counter electrode, whereas an Ag/AgCl (3M KCl) electrode, calibrated with respect to the RHE, was the reference electrode. All potentials reported are referred to the RHE and corrected according to the equation: E(RHE) = E(Ag/AgCl) + 0.218 V + 0.059∙pH. The EC and PEC experiments were carried out in N_2_-saturated 0.5 M HClO_4_ solutions prepared from high-purity reagents and ultrapure Milli-Q water. The electrode was prepared using a Teflon tape to mask the electroactive material, and exposing to the solution just a 0.3 cm diameter circle (area 0.071 cm^2^). The contact was made clamping the sample on a clean part of the DVD, which was kept outside the Teflon tape. Polarization curves were recorded from +0.2 V to −0.5 V vs. RHE, using a scan rate of 5 mV s^−1^. All *V-t* curves were recorded at overpotential (*η*) equal to 0.18 V. Curves were *iR*-corrected, using the resistance found in Electrochemical Impedance Spectroscopy (EIS), which was measured at an overpotential of *η* = 0.18 V. The PEC-HER enhancement was quantified by measuring polarization curves under light illumination from a white neutral light emitting diode (LED, Metrohm, Autolab LED Driver kit), with a light intensity of 97 mW cm^−2^. The transient current measurement was conducted by applying constant cathodic potential (−0.18 V vs. RHE) to the working electrode, and the current response was measured under chopped light from a white LED. Chronoamperometric measurements were performed in 1 h or overnight measurements, fixing a cathodic current density of 10 mA cm^−2^ with light-on light-off cycles of 2 min.

## 3. Results and Discussion

The rapid and easy (30 s, −1.0 V vs. Ag/AgCl) potentiostatic deposition from a 50 mM (NH_4_)_2_MoS_4_ water solution over a Ag-exposed Digital Versatile Disc (DVD) yielded, as previously reported [1], the formation of a heterostructure containing two different sulphides on the metallic surface, namely MoS_2_/Ag_2_S/Ag-DVD nanocomposite. The formation of MoS_2_ is well evident from the XPS analysis of the sample, as reported in Figure 2a (lower spectrum), since the S 2p core level region can be fitted by a couple of doublets. The first one appeared at a binding energy (BE) of 160.9 eV, which is in total agreement with the presence of Ag_2_S [29]; then, a second doublet is found at 161.5 eV of BE, in the typical position of MoS_2_ (for both cases, the position of the doublet is referred to the S 2p_3/2_ component). From the S 2p region, it is possible to define the ratio of MoS_2_ and Ag_2_S, which accounts for 31% and 69% of the total signal, respectively. In addition, some surface Mo oxides, which represents the 49% with respect to the total Mo amount, could be detected in the Mo 3d core level region of the spectrum, as shown in Figure 2a (position of the peaks are referred to the Mo 3d_5/2_ component). These metallic oxides merged out at higher BE (above 231.0 eV of BE), while the component ascribed to MoS_2_ was observed at 228.6 eV of BE (blue curve in Figure 2a) [30]. Indeed, the position of the Mo 3d peak is slightly shifted to lower BE, compared to the typical data of MoS_2_, as a result of the p-n junction formed between MoS_2_ and Ag_2_S [1]. No other element (other than adventitious C and O) could be detected in the XPS survey spectrum. Finally, the formation of a rough layer of MoS_2_ mixed with Ag_2_S and some amount of molybdenum oxides (in agreement with the XPS surface characterization) over the metallic surface of the DVD was imaged by SEM micrographs. (Figure 2c).

Once the presence of metallic sulphides was demonstrated, the MoS_2_/Ag_2_S/Ag nanocomposite was conjugated with thiol-terminated porphyrins **1** and **2**. Briefly, a piece of DVD containing both sulphides was dipped in a 0.02 M ethanol solution of each corresponding molecule under dark and at room temperature for 2 days. Then, it was pulled out and after washing, samples **1-**MoS_2_/Ag_2_S/Ag-DVD and **2-**MoS_2_/Ag_2_S/Ag-DVD were yielded (see Figure 1 and experimental section for details). To shed light on the nature of the interactions between the modified DVD surface and the porphyrin, the hybrid materials were analysed again by XPS (Figure 2). No modification of the Mo 3d or Ag 3d (see Appendix A) XPS core level regions could be detected. We then concluded that the two metals were not involved in the establishment of chemical bonding with the macrocycles. On the other hand, the S 2p core level region of the hybrid samples is rich of information about the porphyrin’s interaction with the MoS_2_/Ag_2_S/Ag-DVD composite surface. In fact, the S 2p region became broader after the wet-chemistry conjugation, and new peaks besides those associated to MoS_2_ and Ag_2_S can be extracted from the fitting procedure. Those new features are even visible by a simple visual inspection of the peak shape. In fact, it is immediately evident that the S 2p signal of sample **1**-MoS_2_/Ag_2_S/Ag-DVD (Figure 2 upper panel) owns a shoulder at higher BE, compared to the pristine nanocomposite. Indeed, this shoulder exactly corresponds with the S 2p signal of the porphyrin deposited on glassy carbon by drop-casting (**1**-GC, Figure 2b medium panel). This S 2p signal is centered at about 163.8 eV, and it is therefore assigned to -SH groups unbound or physisorbed on the surface [31]. Comparing the S 2p signal of the porphyrin-free MoS_2_/Ag_2_S/Ag-DVD surface (lower curve) with that of **1**-MoS_2_/Ag_2_S/Ag-DVD sample (both materials came from the same batch), one realizes that the Ag_2_S/MoS_2_ ratio has changed, with the MoS_2_ signal being apparently more intense in the treated sample. Since any change of this ratio has no physical meaning, it has to be the consequence of the conjugation procedure. Indeed, thiol groups are usually found at a BE of about 161.5 eV [32], which exactly overlaps with the S 2p signal corresponding to MoS_2_. Therefore, the S 2p signal for the sample **1-**MoS_2_/Ag_2_S/Ag-DVD can be properly fitted with a more intense component at 161.5 eV, which suggests the formation of S-S bonds between molecule and surface. Similar observations could be extracted from the S 2p core level analysis of sample **2-**MoS_2_/Ag_2_S/Ag-DVD (Appendix A), highlighting the generality of the protocol. This is confirmed after acquiring the XPS data of Ag_2_S/Ag-DVD sample modified with the porphyrin **1** (Figure 2b), which clearly shows the presence of the component at 161.5 eV (in Appendix A we report the XPS spectra of the as prepared Ag_2_S-Ag-DVD). Even in this sample, the Ag 3d XPS core level region is found identical to the pristine material, suggesting that only the sulphides are interacting the thiols. Although our characterization analysis might indicate that some molecules are physisorbed on the surface of the nanocomposite, the photoemission data also suggests that a fraction of the thiol-terminated macrocycles have been covalently anchored to the material surface, by means of disulphide bridges. Similar results were reported very recently regarding the conjugation of chemically exfoliated MoSe_2_ with thiolated porphyrins [25]. Finally, neither morphological change nor residue were observed by SEM microscopy comparing the pristine and functionalized materials (Appendix A), confirming that the mild conjugation procedure is not affecting the structure.

Vibrational spectroscopies afforded additional valuable information regarding the functionalization process that complements and corroborates the XPS analysis. On the one hand, Raman spectroscopy confirms the amorphous nature of the self-electrosynthetized MoS_2_/Ag_2_S composite, accompanied by the presence of oxides, as reported before [1]. Very broad peaks with low intensity were routinely observed, being centered around, 200, 400 cm^−1^ (bands corresponding to Ag_2_S and amorphous MoS_2_) and 900 cm^−1^ (MoO_x_) (Figure 3a). A more detailed Raman spectrum of the pristine material is presented in the supporting information (Appendix A), together with the Raman spectra of MoS_2_ electrodeposited on FTO and only Ag_2_S grown of DVD. Comparing again the pristine DVD-nanocomposite sample with the functionalized one, new peaks can be observed from the Raman spectra (Figure 3a, red, green). For the two different porphyrins employed, broad peaks in the region of 1300–1600 cm^−1^ of Raman shift were clearly identified. These bands, which were not present in the pristine sample, can be assigned to vibrations concerning carbon bonds corresponding to the organic architectures that can be detected all over the hybrid DVD surface. Similar results concerning the Raman analysis of decorated TMDCs were reported previously [25]. Furthermore, the bands of the metallic sulphides are still present, in sharp agreement with the SEM imaging concerning the minimum structural modification of the sample. In addition, vibrational features were also observed in the ATR-IR spectra (Figure 3b). On this occasion, the peaks corresponding, for instance, to the C-H (aliphatic alkyl chains) and C-S (thiols) vibrations of the organic macrocycle—detected at around 3000 cm^−1^ and 700 cm^−1^ (region highlighted in yellow) of the wavenumber, respectively—were also observed in both conjugated materials. Other bands corresponding to the specific molecule vibrations, for example in the 1500–1750 cm^−1^ region, can also be visible. In stark contrast, the pristine MoS_2_/Ag_2_S/Ag-DVD nanocomposite was silent in these particular spectrum zones. As a whole, vibrational spectroscopies fairly complement the XPS data, supporting the fact that the thiol-terminated porphyrins can be easily deposited on the surface of the DVD by establishing covalent interactions.

For the investigation of the catalytic behavior of our hybrid TMDCs materials toward HER, linear sweep voltammetries (LSV) were performed (Figure 4a). This electrochemical tool has been usually employed in the literature as a functionalization test [7,26], since the sites available for functionalization on the surface of TMDCs are generally the sites responsible for the catalytic activity in the HER too. Therefore, one should expect lower catalytic behavior with the hybrid materials compared to the pristine surface of the MoS_2_/Ag_2_S/Ag-DVD nanocomposite. Measurements performed in 0.5 M HClO_4_ electrolyte showed that the pristine material exhibited an excellent performance towards the HER, presenting an overpotential *η* required to reach a current of −10 mA cm^−2^ of 267 mV vs. RHE after *iR* correction. This performance sharply matches the activity developed by this material previously [1], highlighting this MoS_2_/Ag_2_S/Ag nanocomposite obtained from a DVD as a promising electrocatalyst, considering the amorphous nature and the wide availability of the precursor materials. It is important to mention the sample Ag_2_S/Ag-DVD displayed a much worse HER performance [1], and therefore it was not submitted to the PEC study. Furthermore, the achieved level of activity lies in the range of the state-of-the-art HER electrocatalysts (see Appendix A in the Supporting Information). Surprisingly, the HER behavior of sample **1-**MoS_2_/Ag_2_S/Ag-DVD was slightly better than that of the pristine material, because the overpotential to reach the 10 mA cm^−2^ current was spotted at 264 mV vs. RHE. Furthermore, both samples owned very similar Tafel slopes, indicating that the mechanism of the HER is preserved, following a mixed Volmer-Heyrovsky model (Figure 4b) [33]. This improvement that maintains the mechanism, or lack of prevention, must be caused by the chemical conjugation. Since the chemical structure of porphyrin **1** owns four hydrophilic OH groups, and the functionalization sites and the active sites are supposed to be sterically close, the porphyrin might therefore act as a proton relay [34,35,36], concentrating the active species in a region of the space near the catalytic sites. Hence, the necessary potential to produce hydrogen decreases, determining a better HER activity, especially at low overpotentials. As expected, the HER activity concerning the sample **2-**MoS_2_/Ag_2_S/Ag-DVD was fully blocked, in accordance with the literature data and the hydrophobic nature of this porphyrin. A further proof of this general observation is given by the capacity measurements that have been acquired in the non-faradic region (the complete set of cyclic voltammogram (CV) cycles is reported in Appendix A). In Figure 4c, we report the capacity for MoS_2_/Ag_2_S/Ag-DVD and the same surface coated with porphyrins **1** and **2.** It is very clear that the capacity of **1**-MoS_2_/Ag_2_S/Ag-DVD is about three times that of the sample **2-**MoS_2_/Ag_2_S/Ag-DVD, which confirms a much higher charge density in the double layer caused by the hydrophilic character of the porphyrin **1**. The negative effect of the porphyrin **2** is shown also in the EIS measurements (Figure 4d), collected in dark conditions at −200 mV vs. RHE. While the pristine sample and the one decorated with porphyrin **1** present a nice circle in the Nyquist plot with a very similar charge transfer resistance, the sample modified with the molecule **2** shows a bigger and incomplete circle, at higher frequencies, and in the lower frequencies region the sample behavior is limited by the diffusion of the ions towards the surface.

When illuminated, the samples become even more active. The p-n junction of the pristine MoS_2_/Ag_2_S/Ag nanocomposite, as reported previously [1], has an important role in making this material an efficient photoelectrocatalyst for the HER process. Indeed, illumination with a white LED produced an enhancement of the activity. The overpotential required to reach the onset current of −10 mA cm^−2^ was found to be 262 mV vs. RHE. Nevertheless, the overpotential required to develop the onset current for the hybridized sample **1-**MoS_2_/Ag_2_S/Ag-DVD was found at 257 mV vs. RHE, which represents a 20% of relative light enhancement as a consequence of the functionalization compared to the pristine MoS_2_/Ag_2_S/Ag-DVD nanocomposite. This enhancement represents an absolute decrease of 3% overpotential required to reach the onset current. Since the Tafel slopes are essentially maintained during illumination, we think that the HER mechanism is preserved, and the porphyrin is therefore transferring the excited photoelectron upon illumination, to assist in the HER process, which is strongly supported by the lower charge transfer resistance and higher capacity of the **1-**MoS_2_/Ag_2_S/Ag-DVD functionalized sample (Figure 4d). Furthermore, a stability test by means of chronoamperometry during 60 min of operation at the potential required to develop the onset current (Figure 5a) did not reveal any feature that could be associated to a structural failure, or detachment of the covalently bonded molecule (see Figure 5a). Conversely, stable curve under chopped light was continuously registered for the duration of the experiment. The same behavior was observed in longer measurements, despite the formation of H_2_ bubbles (see Appendix A). It is interesting to note in the case of the sample coated with porphyrin **1** at the same applied bias, where the current density is about 0.4 mA cm^−2^ lower compared to the untreated sample, which corresponds to an improvement of about 15%, in agreement with the LSV experiments. Therefore, the energy position of the frontier orbitals of the porphyrin **1** (HOMO = −5.38 eV, LUMO = −3.51 eV, position referred to the vacuum level) and porphyrin **2** (HOMO = −4.76 eV, LUMO = −2.94 eV, position referred to the vacuum level) [25] allows electrons to be injected into the conduction bands of MoS_2_ and further Ag_2_S upon light excitation [4,37,38,39], thus assisting in the electronic cycle of the already created p-n junction present in the pristine material (Figure 5b). This similar light-induced activation of the catalytic process using porphyrin motifs has been reported in the literature that also employs non-covalent immobilization techniques [40,41]. As a whole, the presence of the porphyrin covalently bonded to the MoS_2_/Ag_2_S/Ag-DVD nanocomposite seems beneficial during the HER process, both under dark or illuminated conditions.

## 4. Conclusions

In this work, we have prepared an efficient material for HER from commercial DVDs, applying a very fast and easy procedure of in two steps. During the first step, we obtained the EC deposition of amorphous MoS_2_ and the concomitant formation of Ag_2_S and MoO_x_ nanoparticles, due to the high concentration of HS^-^ ions in the solution, as result of the reduction and partial hydrolysis of MoS_4_^2−^ ions. In the second step, this surface consisting of a mixture of amorphous MoS_2_ and Ag_2_S was functionalized by two different porphyrins. As a result of the interactions between the S-exposed DVD surface and the thiol-terminated porphyrins, the latter are believed to decorate the nanocomposite by chemical interactions, in particular through disulphide S-S bridges, without interacting with the metals, as the control experiments determined. Regarding the application, we obtained excellent photoelectrocatalysts, considering the facile and rapid synthesis of a common waste product, since the hydrogen production overpotential in perchloric acid was 260 mV in the case of MoS_2_/Ag_2_S/Ag-DVD surface. This value can be further improved if the light is turned on, in the case of sample modified with porphyrin **1**, under illumination, the shifting is enhanced by 20% (value referred to the current density, with respect the pristine material). The use of porphyrin **1** not only makes it possible to have a slight improvement in the overpotentials (both under dark conditions, due to a possible local proton relay phenomena, or illumination conditions), due to the formation of the heterojunction between the porphyrin and the substrate, but also makes it possible to strongly increase the hydrophilicity and the local concentration of protons near the surface. Even though there is room for optimizing the performance enhancement of the final hybrid material, the present work demonstrates that using thiolated photoactive molecules can be advantageous for providing a local proton relay that can favor HER.

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
