# Peer review of "A DVD-MoS2/Ag2S/Ag Nanocomposite Thiol-Conjugated with Porphyrins for an Enhanced Light-Mediated Hydrogen Evolution Reaction"

_nanomaterials, 2020, doi:10.3390/nano10071266_

Round 1

Reviewer 1 Report

This paper reports photocatalytic HER on a DVD-MoS2/Ag2S/Ag nanocomposite thiol-conjugated with porphyrins. Reviewer suggests a major revision before the acceptance by Nanomaterials.

  1. The fitting lines in Fig. 4c and Fig. S4 should cross (0,0) point.
  2. Performance of electrode needs to be prepared with that of state-of-the-art electrodes.
  3. How is the stability of electrode in several hour scale?
  4. English level needs to be improved.

Reviewer 2 Report

The authors report enhancement of the photoelectrochemical H2 evolution by incorporation of porphyrins into the MoS2/Ag2S catalyst on Ag of the DVD.

There are a couple of practical issues that should be considered in the revised version:

  1. The thin 1 um layer of the Ag on the DVD makes it highly resistant, especially if larger areas are utilized, leading to unoptimal current distribution. The resistance are already somewhat large for the small electrode. Are there any suggestions to solve this issue? How was the contact to the DVD made? 
  2. The improvements due to the light illumination are negligible. The light energy was 97 mW/cm2. The electrical energy at 10 mA/cm2 assuming 2 V cell voltage for water splitting would be 20 mW/cm2, i.e. 5 times as much light energy is required. The overpotential improves by 7 mV (from 264 to 257 mV) with porphyrin, and 5 mV without the porphyrin. 20% improvement is a bit misleading, as the overpotential decreased by 2 to 3 %. These overpotentials are still high compared to for example some other non-precious materials such as capsulated iron on CNTs requiring 77 mV to reach 10 mA/cm2. (https://doi.org/10.1002/anie.201411450) 

Round 2

Reviewer 1 Report

Reviewer suggests acceptance of this manuscript.